# Determining the Optimum Level of Soil Olsen Phosphorus and Phosphorus Fertilizer Application for High Phosphorus-Use Efficiency in *Zea mays* L. in Black Soil

Khalid Ibrahim [1,2], Qiong Wang [1], Le Wang [1], Weiwei Zhang [1], Chang Peng [3] and Shuxiang Zhang [1,*]

[1] Institute of Agricultural Resources and Regional Planning, Chinese Academy of Agricultural Sciences/National Engineering Laboratory for Improving Quality of Arable Land, Beijing 100081, China; monzerkhalid442@gmail.com (K.I.); wqcaas@gmail.com (Q.W.); lewangcaas@outlook.com (L.W.); zwwkycg@163.com (W.Z.)

[2] Agricultural Research Corporation (ARC), Wad Medani 126, Sudan

[3] Institute of Agricultural Resources and Environment, Jilin Academy of Agricultural Sciences (Northeast Agricultural Research Center of China), Changchun 130033, China; 391156367@163.com

\* Correspondence: zhangshuxiang@caas.cn

**Abstract:** Phosphorus is an essential macronutrient, both as a component of several important plant structural compounds and as a catalyst in the conversion of numerous important biochemical reactions in plants. The soil Olsen P (OP) level is an important factor affecting crop production and P-use efficiency (PUE). We tested the effect of six OP levels and P doses on maize yield, where the P doses were 0, 22, 44, 59, 73, and 117 kg $P_2O_5$ ha$^{-1}$, with three replications, from 2017 to 2019. The response of crop yield to the OP level can be divided into two parts, below 28 mg kg$^{-1}$ and above 28 mg kg$^{-1}$. The change point between the two parts was determined as the agronomic critical level for maize crops in the study area. The PUE (%) increased with soil OP levels and decreased with P fertilizer application rates. In addition, results for the low P application rate (P2), 22 kg $P_2O_5$ ha$^{-1}$, showed that PUE significantly increased with an increase in the soil OP level compared with PUE at a low OP level (OP1), 0 kg $P_2O_5$ ha$^{-1}$. The PUE value increased by 49.5%, 40.1%, and 32.4% at a high OP level (OP6) in 2017, 2018, and 2019, respectively, compared to that at a low OP level (OP1). At the same OP levels, in all three years, the PUE at a high P application rate (P6) decreased significantly, in the range of 62.8% to 78.7%, compared to that at a low P application rate (P2). Under an average deficit of 100 kg ha$^{-1}$ P, the OP level of the soil in all three years decreased by 3.9 mg kg$^{-1}$ in the treatment without P addition (P1) and increased by 2.4–3.5 mg kg$^{-1}$ in the P treatments for each 100 kg ha$^{-1}$ P surplus. A phosphorus application rate of 44 kg $P_2O_5$ ha$^{-1}$ and an OP level of 28 mg kg$^{-1}$ are sufficient to obtain an optimum yield, increase the PUE, and reduce environmental hazards in the study area in northeastern China.

**Keywords:** phosphorus; relative yield; phosphorus-use efficiency; *Zea mays* L.

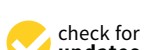



## 1. Introduction

Phosphorus (P) is one of the three essential macronutrients (with nitrogen (N) and potassium (K)) for plant growth and development, and accounts for about 0.2% of the dry weight of the plant [1,2]. Phosphorus is generally considered the main driving force of water eutrophication [3]. Maintaining optimal OP with optimum P application is highly important to reduce water pollution, particularly in hilly areas [4]. Better management of P fertilizer in cropping systems can be achieved by maintaining an optimal balance of input and output of P [5,6]. In China, P fertilizer use increased 91-fold from 1960 to 2008 [7,8], and total grain production increased from 110 million Mg to 483 million Mg, or 3.4-fold, during the same period [9]. Aulakh et al. [10] showed that 26 kg P ha$^{-1}$ is sufficient in a wheat and soybean cropping system that yields 6.55 t ha$^{-1}$ grain yield. A study of a maize and wheat cropping system grown on Aridisol soil reported that the critical P rate

was 40 kg P ha$^{-1}$ for maize yield and 53 kg P ha$^{-1}$ for wheat yield [11,12]. Thus, soil OP levels have improved significantly in most regions of the world [13–15]. Olsen P is the P value extracted from Olsen solution [16], which is used globally to characterize the content of available P in soil. Phosphorus fertilization is a common practice to ensure adequate P supply, leading to extreme soil environmental problems, e.g., leaching, soil acidification, and reduction in soil fertility [17]. Soil should contain the optimum level of OP, in addition to other elements, to ensure the ideal yield and reduce P loss to protect the environment [5,18,19].

The OP level is the main factor influencing crop yield and P-use efficiency (PUE) [20]. Determination of critical levels of OP is important to improve PUE and maximize agronomic yields. The critical level of soil OP for crop production has been defined as the level below which crop production shows a large response to application of P fertilizer, above which the response is minimal. Below the critical level of OP, the crop can no longer absorb sufficient P for growth and severe yield reductions are observed [21]. Chen et al. (2015) [22] found that the critical level of OP and the P rate for maize grain yield in Luvic Xerosols in Sichuan Province in China were 19.1 mg kg$^{-1}$ and 32 kg P ha$^{-1}$, respectively, and the grain yield was 6.3 t ha$^{-1}$. Critical levels of soil OP for maize range between 4 and 15.0 mg P kg$^{-1}$ in alluvial soil in south-western France [23], and the mean critical levels of OP for maize using three models ranged between 12.1 and17.3 mg kg$^{-1}$ in Calcaric Cambisol in China [24]. The critical level of OP usually depends on the crop type [23] and location, due to differences in soil characteristics and climate [25].

The critical level also depends on the target production grade and the availability of other elements [26]. In China, the total net P application is 242 kg P ha$^{-1}$, and the soil OP has increased from 7.4 to 24.7 mg kg$^{-1}$ [5]. Furthermore, the PUE value is below 20% and ranges between 20 and 40 mg kg$^{-1}$ in over 90% of Chinese arable land, and soil OP is above 40 mg kg$^{-1}$ in 9.4% of arable land [12,27]. OP values exceeding 40 mg kg$^{-1}$ pose a threat to the environment [5,28]. Hence, it is important to maintain the optimal yield and properly manage the optimal OP level to reduce the risk of environmental pollution [29]. Scientists are making considerable efforts to determine the critical OP levels that can be used to ensure both optimal crop yields and environmental quality [30,31].

Phosphorus-use efficiency is influenced by several factors, such as OP content, P application rate, soil characteristics, and climate [32–34]. Numerous short-term field experiments conducted around 2000 in China showed that the average PUE of wheat, maize, rice, and other cereal crops was less than 20% [20,35]. Syers et al. [36] noted that the PUE of crops differed in the range between 10% and 50%, according to data from different soils in China, the United States, Brazil, the United Kingdom, India, and Canada [37–41].

To achieve high maize yields in the study area, the PUE needs to be improved as a function of OP in soil with a similar pH, organic matter, and soil texture. Based on field experiments, we hypothesized that optimal application of P fertilizer in maize cropping systems can maximize yield and improve the PUE based on different levels of OP in soil. The objective of this study was to investigate the relationship between PUE and OP, so as to manage P fertilization, optimize the OP level and thus increase PUE in the maize cropping system, and reduce P leaching, which is a major source of pollution.

## 2. Material and Methods

### 2.1. Site Description

The field trials were carried out in 2017, 2018, and 2019 at the Gongzhuling Experimental field in Jilin province, northeastern China (43.5047° N, 124.8228° E). The study site is located in a warm temperate zone with a semi-humid climate. The monthly mean rainfall and temperature values during the cultivation period showed the lowest rainfall (21 mm) and temperature (8 °C) in October, and the highest temperature (24 °C) and rainfall (366 mm) in July and August, respectively. The highest and lowest total seasonal rainfall values were 1074 and 461 mm/year in 2018 and 2017, respectively, as shown in Figure 1. The study area soil is classified as China black soil and Luvic Phaeozems, according to

the FAO system classification [42], and the soil texture is clay loam (clay content 32%) as shown in Table 1.

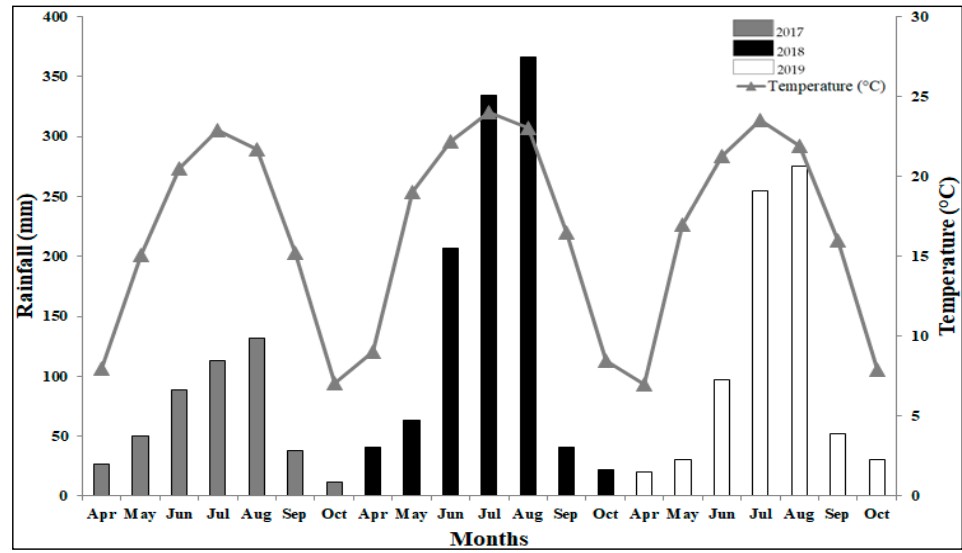

**Figure 1.** Monthly average rainfall (mm) and temperature (°C) at the experimental area in 2017, 2018, and 2019.

**Table 1.** The initial soil characteristics (data are the means from 2017, 2018, and 2019).

| Parameter | Value |
|---|---|
| Organic matter (g kg$^{-1}$) | 22.8 |
| Total N (g N kg$^{-1}$) | 1.40 |
| Total P (g P kg$^{-1}$) | 0.6 |
| Total K (g K kg$^{-1}$) | 18.42 |
| Available P (mg kg$^{-1}$) | 32.3 |
| Soil pH (soil: water = 1:2.5) | 5.6 |
| Bulk density (g cm$^{-3}$) | 1.2 |
| Clay content (<0.002 mm, %) | 32.1 |

*2.2. Experiment Design and Crop Management*

Before planting, soil samples were collected from the experimental site at a depth of 0–20 cm to analyze the initial soil properties for 2017, 2018, and 2019, particularly the initial OP levels and their susceptibility to different doses of P fertilizers in previous years. The field was ploughed by disc harrow during the first week of April. The cropping system was continuously maize (*Zea mays* L.). Maize was planted in late April; the intra-row and inter-row spacings were 25 cm and 75 cm, respectively. Manual weeding was performed to control weeds, and herbicides like (atrazine) were applied during crop growth as needed. The maize was harvested at the end of September. A randomized complete block design was used in the experiment with three replicates, with three main plots. The plot size was $15 \times 4.2$ m$^2$; each plot had 6 rows and the plant density was 53,333 plants ha$^{-1}$. Levels of OP in the plots were dependent on different historical doses of P fertilizer application. We selected six OP levels, marked as OP1, OP2, OP3, OP4, OP5, and OP6, corresponding to 16, 20, 28, 38, 43, and 49 mg kg$^{-1}$, respectively, and the interaction of these OP levels with different rates of P fertilizer application. The P source was P pentoxide (P$_2$O$_5$), namely, P1, P2, P3, P4, P5, and P6, corresponding to 0, 22, 44, 59, 73, and 117 kg P$_2$O$_5$ ha$^{-1}$, respectively. Nitrogen (N) was applied as urea (CH$_4$N$_2$O) at 100 kg N ha$^{-1}$ and potassium as potassium chloride (KCl) at 100 kg K$_2$O ha$^{-1}$ for all treatments. All of the fertilizer treatments were added to the soil before maize planting.

### 2.3. Soil Sampling and Analyses

Soil samples were collected before planting and at harvest time from the plough layer (0–20 cm) during the study period. Fresh soil samples were mixed thoroughly, air-dried, and sieved through a 2.0 mm sieve and stored for analysis. Soil samples were analyzed for soil organic C [43], total P [44], available P [16], and pH in a 1:2.5 soil-to-water ratio [45]. Total N was determined using micro-Kjeldahl digestion, available N (alkaline diffusion method), and 1 mol $L^{-1}$ $NH_4OAc$-extractable K, following the methods of [46].

### 2.4. Plant Sampling and Analyses

The grain yield and aboveground biomass (leaves and stems) of maize were measured at harvesting. Three maize plants were collected from the middle strip of every plot. The samples were air-dried and ground for further analyses. The plant samples were digested with concentrated $H_2SO_4$ and $H_2O_2$ (30%) and then the total P concentration was estimated using the molybdate method [47]. The phosphorus concentration was multiplied by grain yield and aboveground biomass yield to calculate the P absorbed by the grain and aboveground biomass, and then the phosphorus-use efficiency (PUE) was calculated:

$$PUE = \frac{U_A - U_0}{P_A} \times 100 \qquad (1)$$

where $U_A$ is the amount of P absorbed by the maize treated with P, $U_0$ is the amount of P absorbed by crops without P treatment, and $P_A$ is the amount of phosphorus applied. Considering that the experiment was carried out in the same location for three consecutive years, we calculated the PUE for the maize crop with the same precept [48].

$$P_B = P_A - P_{gb} \qquad (2)$$

where $P_B$ is the apparent P balance, $P_A$ is the amount of phosphorus applied, and $P_{gb}$ is the total amount of P absorbed by grains and the aboveground biomass of maize crops. The relative yield was calculated to find the relationship between the relative yield under six Olsen P levels in black soil at Gongzhuling in northeastern China during the study period (2017, 2018, and 2019), as follows:

$$Y_R = (Y_f - Y_m) * 100 \qquad (3)$$

where $Y_R$ is the relative yield, $Y_f$ is the yield of different treatments (kg ha$^{-1}$), and $Y_m$ is the maximum yield (kg ha$^{-1}$) among the treatments.

### 2.5. Statistical Analysis

Regression equations for correlations between the soil OP levels and P application rates with grain yield and aboveground biomass yield were determined with Duncan's test using the SAS 9.0 software package (SAS Institute Inc., Cary, NC, USA). Significant differences in PUE and OP level among the various P applications were determined at the $p < 0.05$ significance level using the least significant difference (LSD), using the SPSS 21.0 software package. The figures were drawn using the SigmaPlot 12.5 software package.

## 3. Results

### 3.1. Grain Yield Response to Olsen P Levels and P Application Rates during the Three Years of the Experiment

The P fertilizer treatments and six OP levels had a significant effect on the grain yield of maize. All fertilizer treatments produced a significantly higher grain yield than the control in all three years (2017, 2018, and 2019). We observed that grain yield with P1 (control) gradually increased with the increase in OP level from OP1 to OP3 in all three years and then remained stable in OP4 to OP6, as shown in Figure 2. Comparing the grain yield for the three years 2017, 2018, and 2019, it was observed that the highest grain yield

was recorded in 2018 (12.4 Mg ha$^{-1}$). This is because the average rainfall in 2018 was relatively higher than in 2017 and 2019, as shown in Figure 1.

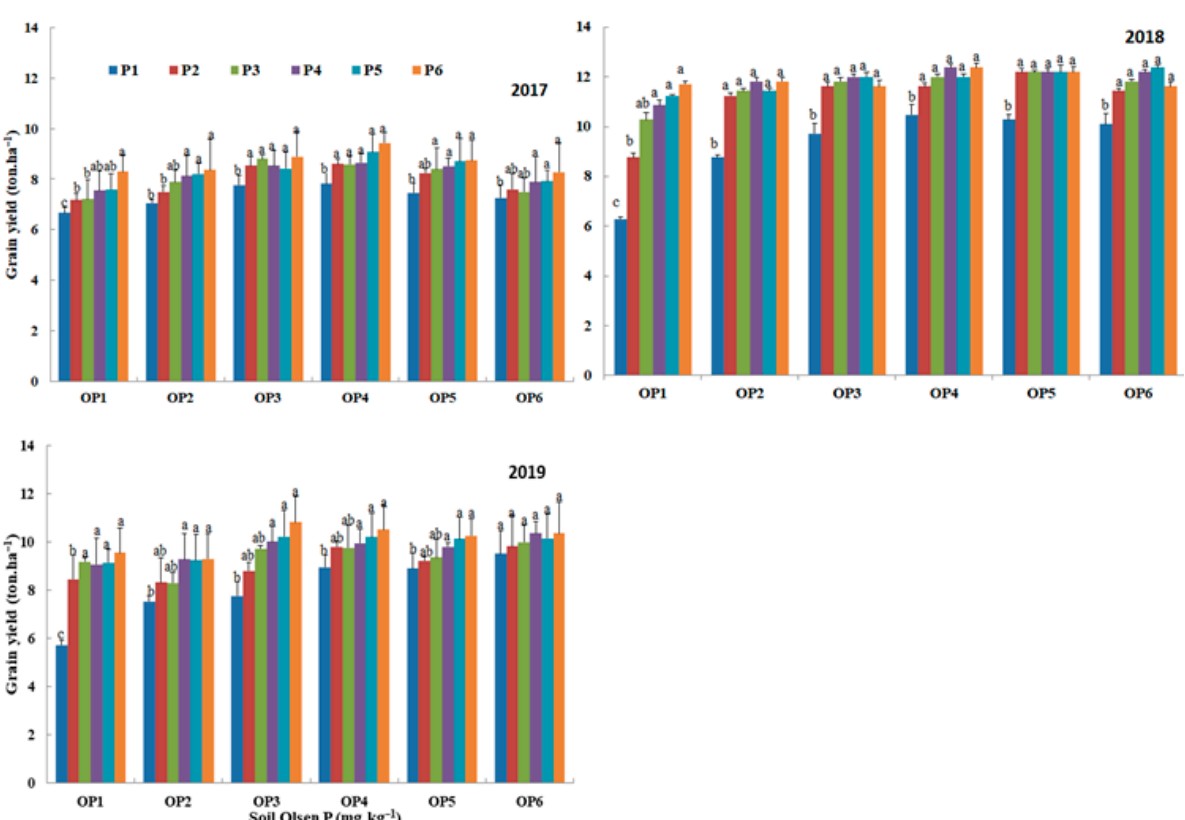

**Figure 2.** The grain yield of maize under six Olsen P levels and P application rates in black soil at Gongzhuling, northeastern China, in 2017, 2018, and 2019. Different letters within a column are significantly different at $p < 0.05$. OP represents the Olsen P levels, OP1, OP2, OP3, OP4, OP5, and OP6, corresponding to 16, 20, 28, 38, 43, and 49 mg kg$^{-1}$, respectively, and P represents the P application rates, P1, P2, P3, P4, P5, and P6, corresponding to 0, 22, 44, 59, 73, and 117 kg P$_2$O$_5$ ha$^{-1}$, respectively.

### 3.2. Grain and Aboveground Biomass Yield Response to Olsen P Levels and P Application Rates

Figure 3a,b show the average grain yield and aboveground biomass of maize crops during the study period (2017, 2018, and 2019) under six OP levels and P application rates. Compared with P1 (control), all P fertilizer treatments significantly increased the grain yield and aboveground biomass during the study period. The grain yield in the P1 treatment increased gradually with the increase in OP levels from OP1 to OP3, and was then stable from OP4 (38 mg kg$^{-1}$) to OP6 (49 mg kg$^{-1}$), although the Olsen P level increased. With the improvement in the soil fertility level, the effect of P fertilizer application on yield was not obvious, and there was no significant difference in yield between various P fertilizer treatments.

Phosphorus fertilization significantly increased the crop production ($p < 0.05$), especially in the treatments with P application, as shown in Figure 3. This shows that P deficiency has a significant effect on the grain yield of maize. In our study, we observed that grain yield and aboveground biomass yield increased by 42.6% and 36.0%, respectively, compared to P1 (control). Specifically, the average grain yield and aboveground biomass were 38.6% and 32.5% higher in the P6 treatment with high OP (OP6) than in the control treatment (P1).

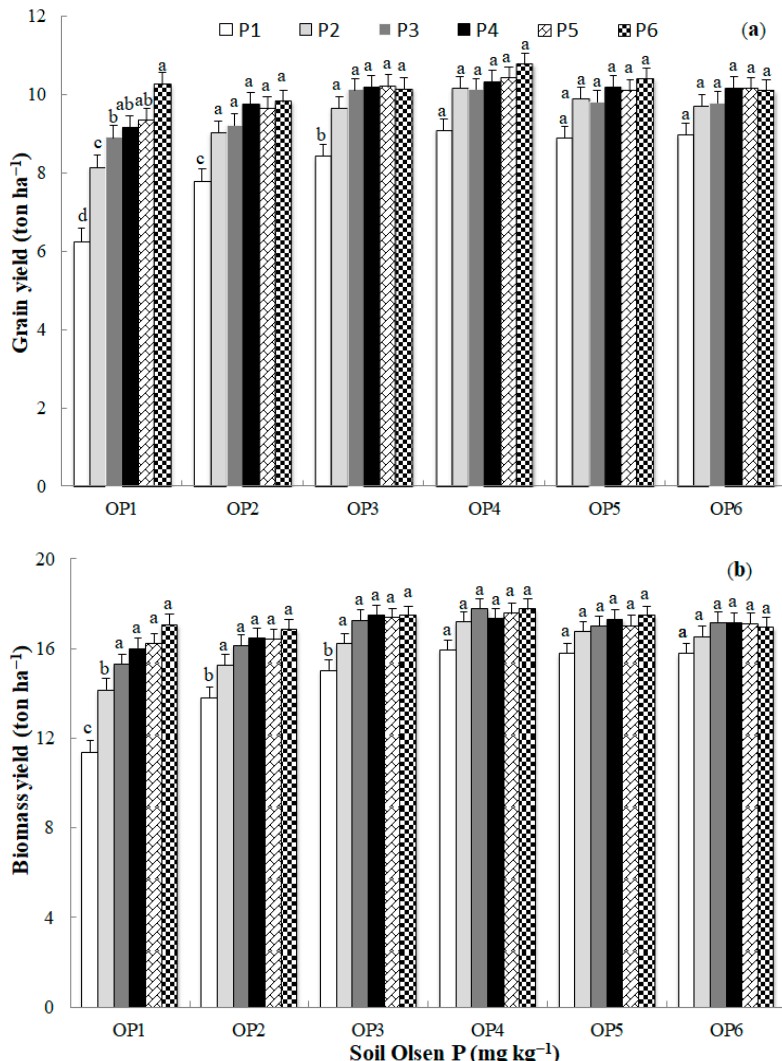

**Figure 3.** (**a**) The average grain yield and (**b**) aboveground biomass yield of maize under six Olsen P levels and P application rates in black soil at Gongzhuling, northeastern China, during the study period 2017, 2018, and 2019. Different lowercase letters indicate a significant difference ($p < 0.05$) between grain and aboveground biomass yield at each Olsen P level. OP represents the Olsen P levels, OP1, OP2, OP3, OP4, OP5, and OP6, corresponding to 16, 20, 28, 38, 43, and 49 mg kg$^{-1}$, respectively, and P represents the P application rates, P1, P2, P3, P4, P5, and P6, corresponding to 0, 22, 44, 59, 73, and 117 kg P$_2$O$_5$ ha$^{-1}$, respectively.

### 3.3. Grain and Aboveground Biomass Yield Response to P Application Rates

Figure 4a,b show the average grain yield and aboveground biomass of maize crops during the study period 2017, 2018, and 2019 under six P application rates. We compared P1 (control) in average grain yield and aboveground biomass (8.1 and 13.5 t ha$^{-1}$) with other P fertilizer application rates, P2, P3, P4, P5, and P6, corresponding to 9.2, 9.8, 10, 9.9, and 9.8 and 14.7, 15.6, 15, 8, 15.8, and 15.9, respectively. The grain and aboveground biomass yield significantly increased during the study period. The grain yield with P1 (control) increased gradually with an increase in P application from P1 to P3, and was then stable from OP4 (59 kg P$_2$O$_5$ ha$^{-1}$) to P6 (117 kg P$_2$O$_5$ ha$^{-1}$), although the P application rate increased.

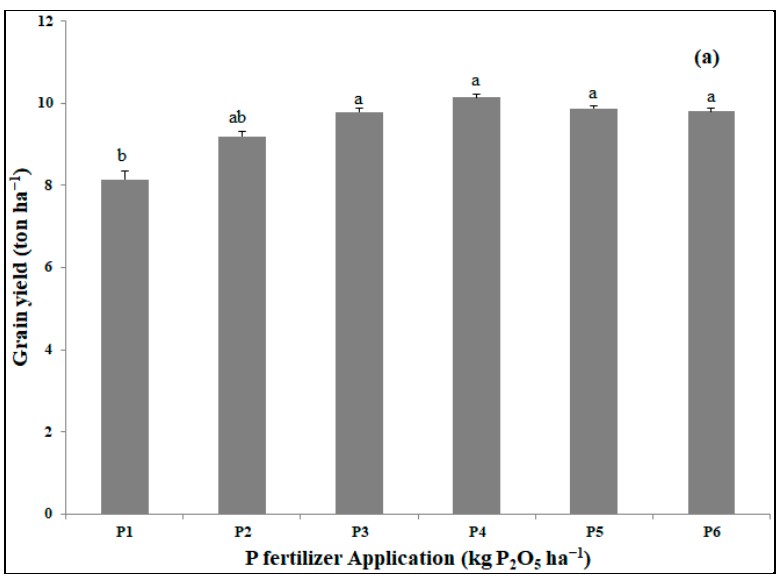

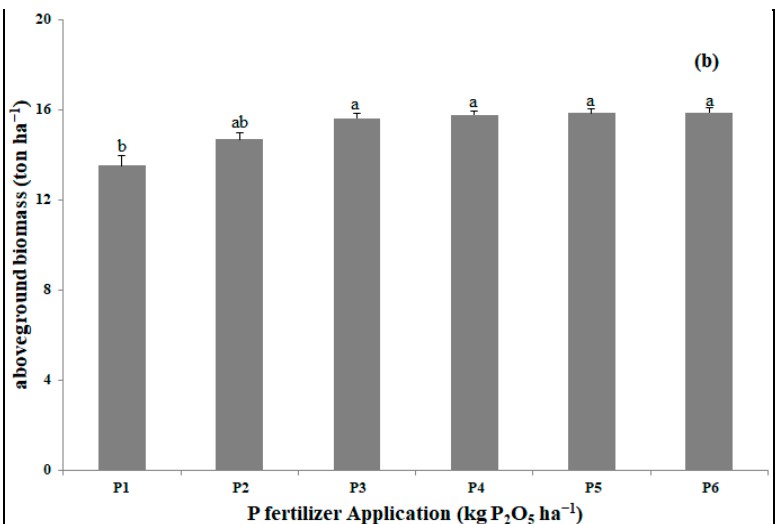

**Figure 4.** (**a**) The average grain yield and (**b**) aboveground biomass yield of maize under P application rates in black soil at Gongzhuling, northeastern China, during the study period 2017, 2018, and 2019. Different lowercase letters indicate a significant difference ($p < 0.05$) between grain and aboveground biomass yield at each of the P rates. The phosphorus fertilizer application rates were P1, P2, P3, P4, P5, and P6, corresponding to 0, 22, 44, 59, 73, and 117 kg $P_2O_5$ ha$^{-1}$, respectively.

### 3.4. Phosphorus-Use Efficiency Response to Soil OP Levels and P Application Rates

Figure 5 shows the PUE (%) by maize crop at six OP levels and P application rates during the study period of 2017, 2018, and 2019. We observed that PUE (%) showed a positive trend with OP levels and a negative trend with P application rates, as shown in Figure 5. When comparing PUE at the low OP level (OP1) with PUE at the high OP level (OP6) at the same P application rate (P2), we observed that the PUE increased significantly with an increase in soil OP level. Phosphorus-use efficiency increased at a high OP level (OP6) in the range of 32.4–49.5% compared to PUE at a low OP level (OP1) during the study period of 2017, 2018, and 2019. During the study period, when the PUE value at the low P application rate (P2) was compared with the PUE value at the high P application rate (P6) at the same OP levels, it was found that the PUE value decreased significantly, in the range of 62.8–78.7%. The maximum PUE during the study period was 33.0% recorded at OP6 at the low P application rate (P2). The minimum PUE was 5.9%, recorded at OP1 at the high P application rate (P6), as shown in Figure 5.

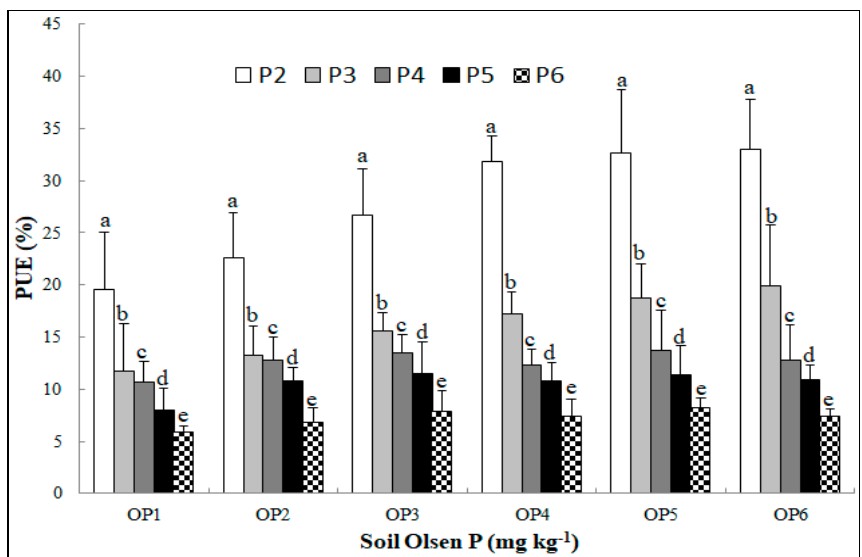

**Figure 5.** The average phosphorus-use efficiency (PUE) of maize crops under six levels of Olsen P and P application rates in black soil at Gongzhuling, northeastern China, during the study period 2017, 2018, and 2019. Note: OP1, OP2, OP3, OP4, OP5, and OP6 refer to the six Olsen P levels of 16, 20, 28, 38, 43 and 49 mg kg$^{-1}$, respectively. P2, P3, P4, P5, and P6 refer to the five phosphorus application rates of 22, 44, 59, 73, and 117 kg P ha$^{-1}$, respectively. Different lowercase letters within a column indicate significant differences at different P fertilizers rates within the Olsen P level ($p < 0.05$).

### 3.5. Phosphorus-Use Efficiency Response to P Application Rates

Figure 6 shows the average PUE (%) of the maize crops at six P application rates during the study period of 2017, 2018, and 2019. We observed that PUE (%) showed a negative trend as the P application rate increased. Comparing the PUE at a low P application rate (P2 22 kg P$_2$O$_5$ ha$^{-1}$) with the PUE at a high P application rate (P6 117 kg P$_2$O$_5$ ha$^{-1}$), we observed that the PUE decreased significantly with the increase in P application rate during the study period. During the study period, the maximum PUE was 28%, recorded at P2, and the minimum PUE was 7%, recorded at P6, as shown in Figure 6.

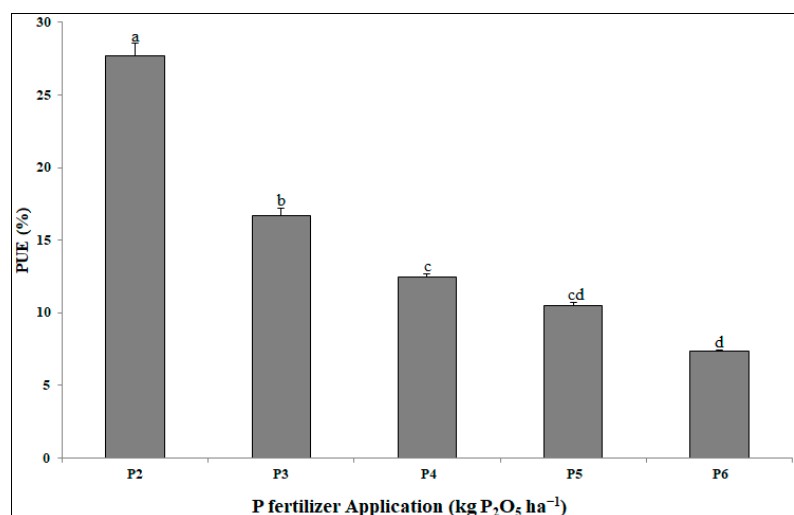

**Figure 6.** The average phosphorus-use efficiency (PUE) of maize crops under six P application rates in black soil at Gongzhuling, northeastern China, during the study period 2017, 2018, and 2019. The phosphorus fertilizer application rates were P1, P2, P3, P4, P5, and P6, corresponding to 0, 22, 44, 59, 73, and 117 kg P$_2$O$_5$ ha$^{-1}$, respectively. Different lowercase letters within a column indicate significant differences at $p < 0.05$.

### 3.6. The Agronomic Critical Level of Olsen P for Maize in Black Soil

The correlation between relative yield and soil OP levels, fitted with the Michaelis equation, is shown in Figure 7. Relative yield increased with an increase in soil OP level. The response of relative yield to the levels of OP was divided into two parts. In the first part, the OP level was below 28 mg kg$^{-1}$, and the yield increased significantly with the increase in OP level and was significantly related to the amount of P fertilizer. In the second part, above 28 mg kg$^{-1}$, there was no response to P fertilizer, so the crop yield was stable despite the increase in OP level. The change point between the two parts was considered to be the agronomical critical level for OP.

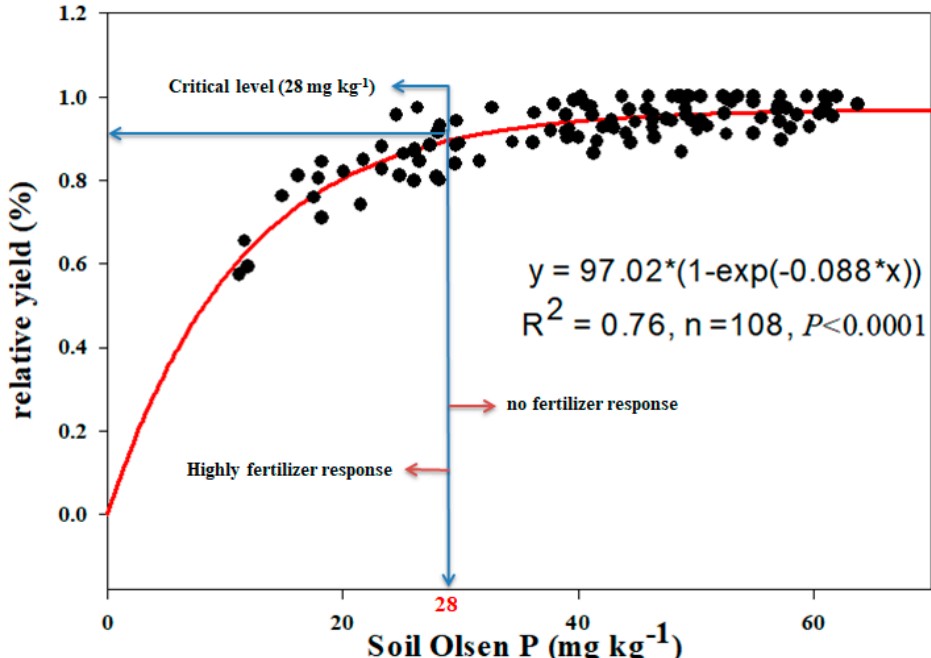

**Figure 7.** The relative yield under six Olsen P levels in black soil at Gongzhuling, northeastern China, during the study period 2017, 2018, and 2019.

### 3.7. Effect of Experimental Factors Year, Olsen P, and P Application Rate on the Growth, Yield, and PUE of Maize Crops

The interaction between the three experimental factors—year, OP, and P application—with grain yield, aboveground biomass, and PUE, is shown in Table 2. The grain yield and aboveground biomass showed a highly significant difference, $R^2$ = 0.97 and 0.62, respectively ($p < 0.01$), with year, OP, P application, and year × OP. Regarding PUE, there was a highly significant difference, $R^2$ = 0.96 ($p < 0.01$), for all the experimental factors, with the exception of year × OP × P, which showed no significant difference, as shown in Table 2. From the results in Table 2, we can conclude that the factor OP is the main factor affecting the maize yield (grain and above ground biomass) and PUE value of the maize plant.

**Table 2.** Summary of statistical analysis of the effect of year, Olsen P, and P application on grain yield, aboveground biomass, and PUE.

| Dependent Factor: Grain Yield | | | |
|---|---|---|---|
| **Factor** | **df** | **Mean Squares** | ***p*-Value** |
| Year (Y) | 2 | 1486.8 | ** |
| Olsen P (OP) | 5 | 2.4 | ** |
| P application (P) | 5 | 15.9 | ** |
| Y × OP | 10 | 1.8 | ** |
| Y × P | 10 | 1.9 | ** |
| OP × P | 25 | 0.8 | * |
| Y ×OP × P | 50 | 0.6 | ns |
| Error | 216 | 0.5 | - |
| $R^2$ = 0.97 | - | - | - |
| Dependent factor: aboveground biomass yield | | | |
| Year | 2 | 117.6 | ** |
| OP | 5 | 16.9 | ** |
| P | 5 | 68.4 | ** |
| Y × OP | 10 | 5.9 | ** |
| Y × P | 10 | 2.3 | ns |
| OP × P | 25 | 2.7 | ns |
| Y × OP × P | 50 | 1.2 | ns |
| Error | 216 | 2.4 | - |
| $R^2$ = 0.62 | - | - | - |
| Dependent factor: phosphorus-use efficiency (PUE) | | | |
| Year | 2 | 1208.4 | ** |
| OP | 5 | 105.0 | ** |
| P | 5 | 4252.4 | ** |
| Y × OP | 10 | 23.7 | ** |
| Y × P | 10 | 109.6 | ** |
| OP × P | 25 | 34.2 | ** |
| Y × OP × P | 50 | 10.0 | ns |
| Error | 216 | 9.9 | - |
| $R^2$ = 0.96 | - | - | - |

Note: ns: no significant at the 0.05 level; * Significant at the 0.05 level; ** Significant at the 0.01 level.

## 4. Discussion

### 4.1. Crop Yield Response to Soil Olsen P Levels and P Application Rates

Excessive P application leads to significant environmental problems [49], whereas insufficient P input will lead to low crop yields and soil fertility degradation [50,51]. Thus, optimal P application is necessary for a sustainable agricultural system. This research supports our hypothesis that optimal P application will lead to higher crop production and improve PUE in the maize cropping system. The critical OP level can be described as the OP level at which the yield does not change despite the increase in OP levels and P application rate. The critical OP level for a continuous maize cropping system in black soil in the study area was 28 mg kg$^{-1}$, as shown in Figure 7. We observed the P application rate, 44 kg P ha$^{-1}$, was sufficient to meet crop requirements for optimal crop yield. Xia et al. [11] noted 40 kg P ha$^{-1}$ was sufficient to meet the maize crop requirements of P in clay soil, where the OP level was 20.3 mg kg$^{-1}$.

Moreover, in this study, we observed that there was no response to yield increases when the OP level reached 28 mg kg$^{-1}$ in all P application rates (P2 to P6). Tang et al. [52] noted that growing crops without P fertilizer application resulted in a significant decrease in crop productivity over time in many locations in China. Continuous cultivation without P fertilizer application significantly reduces the OP levels in the soil. Although we observed a difference in OP level between treatments with P fertilization and treatments without P fertilization, the OP level decreased significantly with continued cultivation

without P fertilization, whereas the OP level increased significantly with continued P fertilization [53,54].

### 4.2. The Phosphorus-Use Efficiency of Maize and Olsen P Influence on Maize Yield

The PUE could be used to characterize the P effects [55,56]. In China, about 15–20% of the P is absorbed by plants in the year of P fertilizer application [27,57]. In our study, among six OP levels and P application treatments, there were significant differences ($p < 0.05$) in PUE during the study period, as shown in Figure 5. In general, the PUE was low in the study area. Due to the low soil pH (5.5 to 6) in this area, some elements are more active, particularly aluminum (Al), which reacts with P to form compounds that lead to P precipitation, thus making P unavailable for the plant.

In our result, we observed the maximum PUE was 33% at a high OP level (OP6) and a low P application rate. The minimum PUE was 5.9% at a low OP level and a high P application rate. The PUE at a high OP level and a low P application rate was 5.6 times more than the PUE at a low OP level and high P application rate. This indicates PUE has a positive correlation with the OP level and a negative relationship with the P application rate. Xu et al. [58] reported that a higher PUE value was obtained using the optimal P fertilizer management practice, whereas high rates of P fertilizer result in a low PUE value, due to the overuse of P fertilizer. Xin et al. [59] and Chuan et al. [60] noted that the average PUE values for maize and wheat in China were 15.7% and 10.2%, respectively. Dobermann et al. [61] suggested that P inputs in grain production systems in soils that are not readily P-fixable should aim to achieve 30 to 50 kg of grain for each kg of P applied. Moreover, the main problem of most soils globally is a low PUE, which is the main reason why PUE needs to be improved [24,62,63]. Overall, significant effort still needs to be made regarding the best management practices to further improve PUE in China.

### 4.3. Critical Levels of Olsen P for Crop Yield

A critical OP level can be defined as "a soil P status above which crop yield does not respond to P fertilization" [24,64,65]. In our study, the critical level of OP for maize was 28 mg kg$^1$. Once the critical level of OP is reached or slightly exceeded, P fertilizer application should be reduced to maintain the existing OP level [25,66]. Tang et al. [24] reported the critical OP level for maize in the range of 13–15 mg kg$^{-1}$. Critical OP levels for maize in the state of Idaho in the western US and France range from 7 to 11 mg kg$^{-1}$ [23,67]. Li et al. [5] noted that the critical OP level for rice in China ranged between 10 and 20 mg kg$^{-1}$, and Bado et al. [68] noted a level of 17 mg kg$^{-1}$ as a critical OP level for rice in Africa. Changes in critical levels may also result from differences in the models used to estimate the critical levels [64].

Critical OP levels range between 10 and 40 mg kg$^{-1}$, depending on the climate, soil properties, and crop type. When the soil OP is below the agronomically critical levels of OP, the PUE increases significantly if the soil OP is increased; when the soil OP level is very close to the agronomically critical levels of OP, the PUE may reach the maximum value or change slightly if the level of the soil OP changes; and when the soil OP level is significantly above the agronomically critical levels of OP, the PUE decreases with increasing OP level. The critical level of OP in our study was 28 mg kg$^{-1}$, as shown in Figure 7, consistent with the range of 10–40 mg kg$^{-1}$ reported by Jordan [69]. This critical OP level allowed us to determine the critical OP level beyond which crop yields do not respond to P application (Figure 7).

### 4.4. Phosphorus Fertilizer Management Strategies

The P management strategy used in agriculture can be based on the critical level of the soil OP, because OP is the most commonly used index in China [64]. By combining the relationships among soil OP, total P, and P budget (P input by fertilizer minus P output by crop uptake) [70], the amount of P fertilizer needed to adjust soil OP to the agronomically critical level can be calculated. Moreover, the PUE should be considered to protect the

limited P resources and reduce P loss risk [71]. In addition, P leaching has been shown to occur in many soil types in China when the soil OP level is higher than 40 mg kg$^{-1}$ [64]. According to the present study, to achieve a relatively high crop yield, PUE, and soil fertility, the optimum OP should range from the agronomically critical level (28 mg kg$^{-1}$) to the leaching change-points of OP (usually 40 mg kg$^{-1}$ for most soils of China). Thus, we inferred that the optimum Olsen P level should be around 28 mg kg$^{-1}$ in the study area in northeastern China.

### 5. Conclusions

A rational P management strategy can contribute to enhanced total grain yields and PUE in maize cropping systems. Strategies used for soil P management are implemented to achieve a balance among food security, resource limitations (high PUE), and pollution prevention in the field. First, the agronomically critical levels of soil OP for crop yield should be achieved. In the current study, the agronomically critical level of OP varied depending on soil types, crop species, and climate. Over time, the PUE presented an increasing trend in the study area, and an initially increasing and then an obviously decreasing trend. Moreover, the correlation was used to describe the response relationship of PUE with soil OP. All of the change rates were very small, indicating that the PUE values increased slightly or remained unchanged as the soil OP increased under most of the fertilization treatments in the study area. When the soil OP level is very high, the PUE decreases, wasting P resources and leading to significant environmental problems, such as the eutrophication of water bodies. Thus, the OP level should be maintained at an optimal level. Based on the data from the three-year period of the field experiment, to achieve a high PUE, crop yield, and soil fertility, the optimal level of soil OP is 28 mg kg$^{-1}$ in the study area. Furthermore, to realize the goal of optimizing P use in diverse soil types and crop systems in China, additional studies and collaborations are required.

**Author Contributions:** Conceptualization, K.I. and S.Z.; writing—original draft preparation, K.I.; writing—review and editing, K.I. and S.Z.; resources and data curation, Q.W. and L.W.; visualization, W.Z. and C.P.; supervision, S.Z.; funding acquisition, S.Z. All authors have read and agreed to the published version of the manuscript.

**Funding:** This work is supported by the National Key Research and Development Program of China (2016YFD0200301), by the Special Fund for Agro-scientific Research in the Public Interest of China (201503120) and by the National Natural Science Foundation of China (41977103, 41471249).

**Institutional Review Board Statement:** Not applicable.

**Informed Consent Statement:** Not applicable.

**Data Availability Statement:** Not applicable.

**Acknowledgments:** We thank all staff for their valuable work associated with these long-term Monitoring Networks of Soil Fertility and Fertilizer Effects in China.

**Conflicts of Interest:** The authors declare no conflict of interest.

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
