# Peer review of "Determining the Optimum Level of Soil Olsen Phosphorus and Phosphorus Fertilizer Application for High Phosphorus-Use Efficiency in Zea mays L. in Black Soil"

_sustainability, doi:10.3390/su13115983_

Round 1

Reviewer 1 Report

The phosphorus (P) fertilization is one of the most important agricultural practices which increases crop production but at the same time significantly affects the environment. P is also a nonrenewable resource, sourced from the phosphate rocks. Therefore the optimal P fertilization is one of key issues in both agriculture and environmental protection. The manuscript entitled “Effect of Soil Olsen P Levels on the Yield of Maize and Phosphorus Use Efficiency in Black Soil” describes the results of investigations on critical level of soil Olsen phosphorus in black soils in China. The Introduction is quite extensive, however very hard to follow. The Material and methods are described poorly. Much basic information is missing. The design of experiment is unclearly described.  The Results are not sufficiently described and are very hard to follow. The relations and effects of different factors are not clearly described and are not always supported by properly designed statistics. The Discussion is quite good; however, does not explain the obtained results. The Conclusions are more like repeating the main results. The extensive English revision of the manuscript is needed.

Below, there are several major comments:

  1. In Abstract the sentences should not be the same as in the main text.
  2. In the Introduction I miss short description of main environmental consequences of P fertilization
  3. In Site description I miss the type of climate, the mean highest and the lowest temperatures in the year, months with the mean highest precipitation from multi-year observation. I miss also an interpretation of differences in precipitation in 3 years of study period. The particle size distribution of soil should be added. This is more accurate information than texture which is different in different countries.
  4. Figure 1 shows only temperature and precipitation from vegetation period. This should be added in the text.
  5. Table 1 should contain only physico-chemical characteristics of the soil. The other information should be given in the description.
  6. In Site description the information about earlier fertilization rate should be given as well as the short history of the field i.e. for how long the maize monoculture has been cultivated.
  7. Why the 50 cm of inter-row spacing was used? Typically it is 75 cm.
  8. What do you mean by “We selected six OP levels”? How did you select those soils? There is an initial soil with OP content 11.9 mg kg-1, so how did you have soils with different OP content? It also should be added in the description that second factor was P fertilization which rates are given in table 2.
  9. In which years the soil and plant material were sampled?
  10. When (month) was maize harvested?
  11. How many plots were in total?
  12. In Soil sampling and analysis (should be rather analyses) all methods should be described shortly and the citation should be given. The equipment used for analysis also should be given. The methods used for analysis of initial soil characteristics also should be given. Was the moisture of soil measured and how? Did you measure the total phosphorus content?
  13. In Plant sampling and analysis only one analysis is described. Was water content or total solids/dry matter of plants measured? Was ash content of plants measured? The description of analysis should be more accurate with citation and equipment. The plant material should be dried in a certain temperature like usually 65-70oC or 105oC to the constant weight. The P concentration measured in plant material is total phosphorus (TP). What do you mean by biomass? Is it whole plant material with grains?
  14. Did you check the results for normal distribution? Since you had two factors affected the two-way ANOVA should be used to analyze the differences in grain yield, biomass yield and PUE and OP content?
  15. The Results are written very unclearly and are hard to follow. The main weakness is the lack of two-way ANOVA which could easily explain most of the relationships between biomass, grain yield, PUE etc. within OP levels and P fertilization rates eg. lines 219-221
  16. There should be clear distinction in OP level and P treatment which means P fertilization rate. This both factors are mixed up in the Results and Discussion.
  17. I miss any information on the results comparison between each of 3 years of experiment. It could be interesting to show those results since 2017 seems to be very dry comparing to two other years. The last paragraph (3.4) is not enough.
  18. I miss short description of grain and biomass yield and TP content in plants.
  19. In Discussion the “reasonable” should be changed to “optimum”.
  20. Is the P fertilizer rate of 44 kg P ha-1 sufficient despite the OP level? This is in contradiction with next sentence. What do you mean by “ P application rate can vary based on yield level”?
  21. What results of Tang et al have in common with critical OP level. Tang et al. observed the depletion of available phosphorus  in unfertilized soils under cultivation and maize yield decrease in areas where soils were not fertilized with P, while critical OP value is related to the yield response to increasing OP levels in soil.
  22. In paragraph 4.2. The Phosphorus Use Efficiency and Olsen P influence in the maize yield there is no explanation why the PUE is so low? Is it only a problem soils? Or is it a problem of plant physiology, or other nutrients?

Below, there are several minor comments:

In whole manuscript I would change the “reasonable application” to rather optimal or rational

In whole manuscript please check spaces and double spaces (eg. lines 46, 51, 52, 68, 71, 83, 156, 207, 235, 256 and others)

In whole manuscript please unify dashes and way of writing numerical ranges with a hyphen.

All figures and tables should be centered on the page.

Line 16: There is no verb in this sentence.

Line 22: there should be “was considered” instead of “considered”

Line 23: this sentence means the same as sentence before

Line 25: there should be “increases” and “reduces”

Line 36: there should be a space between “China” and [5]. This sentence should be checked for grammar.

Line 40: there should be “elements” instead of “the elements”

Line 46: rather “above” instead of “after”

Line 52: “maize utilizing three models” – what does it mean?

Line 53: please check this part of the sentence “The critical level is usually differed because of …”

Line 57: there should be “In China, from 1980…”. If in the sentence, there is a time range there should be information of difference in P application, not only OP increase or the sentence should be changed.

Line 58: please delete comma and full stop before citation.

Line 58: “7.4 mg / kg to 24.7 mg kg-1” – please unify the units

Line 58: the sentence “Meanwhile…” needs checking for grammar

Line 59: “The range of OP between 20- 40 mg kg-1 in the dominant Chinese arable land about 90%” – there is no verb in this sentence

Line 60: please delete space in 20- 40

Line 61: “which the value of the OP that exceeds 40 mg kg-1 poses a threat to the environment” – this part needs a revision

Line 62: perhaps “ Hence, it is important “ would be better than “So, it is meaningful”

Line 63: please delete “where”

Line 66: there should be “like OP content” and “application rate” instead to “application amount”

Line 68: has shown or showed

Line 69: please delete 2008

Lines 72-78: this paragraph should be checked for grammar

Line 95: “and it's affected by different doses of P fertilizers application in the previous period.” – this part of sentence needs revision or/and explanation

Line  96: “the plough” is a farming tool, and “to plough” is to dig the land with a plough. Therefore the sentence should be “The field was ploughed.” Please, specify when the field was ploughed.

Line 98: do you mean “inter-row and intra-row spacings” ? Do you mean intra-row (spacing for plants in the row) – 25 cm and inter-row spacing (spacing between rows) – 50 cm?

Line 100: please delete “measuring”

Line 102: nitrogen and potassium should be written with lower case letters

Table 2: Why P in unit “(kg P ha-1 year-1)”? Nitrogen and potassium should be written with lower case letters. P1 should be described as control.

Line 107: the space is missed in “0-20cm”, maize should be written with small letters

Line 108: What do you mean by that “After harvesting in 2018 and 2018”? Line 110: the sentence needs checking for grammar. It sounds like re-written from manual.

Line 115: middle or central not center strip

Line 116: there should be “and grounded for further analyses.”

Line 116: there should be “and then TP concentration estimated by the molybdate…”

Line 129: This sentence needs a rewriting.

Line 141: please check this “The grain yield with P1 (control)”. I would suggest “ The grain yield in P1 treatment.

Figure 2. The OP1-6 symbols and P1-6 symbols should be explained

Line 155: the sentence should start with capital letter

Line 157: why “Significantly” with capital letter?

Figure 3: phosphorus with lower case letter

Line 170: This sentence should be re-written. The lowercase letters indicate the significant differences among PUE. Is it a difference among PUE at different P fertilizers rates within the Olsen P level or among PUE at the same P rate but different Olsen P level.

Line 173: Full stop is missing.

Lines 175-178: the sentences should be re-written. This is unclear.

Line 175: “could be divided” instead of “could be portioned”

Line 182: the sentence should start with capital letter

Line 187: “is shown” not “as shown”

Line 189-191: the sentence is unclear

Line 199: delete “produce”

Line 203: “despite” instead “although”

Line 208: please delete “to get optimum production”. It is obvious from the sentence.

Line 210: please change “yields 6.55 t ha-1 grain yield” You cannot compare wheat and soybean yield with maize.

Line 212: Why is it critical P value? Isn’t it fertilization rate which meets the crop requirements?

Line 215: I do not understand this sentence. What was the treatment with P fertilization but with no P addition?

Line 217: there should be “maize grain yield”

Line 217: “high maize grain and biomass was 10.8 and 17.8 in the range 6.2-10.8 and 11.4-17.8 ton ha−1,” – what does it mean?

Lines 215-222 belongs to Results

Line 221: perhaps it should be “but they were 6.5 and 5.1 lower than that in treatment P6 with OP4”

Line 229-233: The sentences should be re-written because they are unclear.

Line 245: should be “is called”

Line 248: “that are not easily P fixation” grammar should be checked

Line 249: kg with lower case letter

Line 249: grammar should be checked

Line 259: should be “between 10 and 20”

Line 283: How application rate can effectively absorb nutrients?

Line 284: If the recommended P fertilization rate is much higher than your results how is it possible that there is a balance in agriculture and the outflow of P is low?

Line 290: Maize should be in lower case letters

Line 291: “until when OP levels were increased” –what do you mean by that?

Line 293: The first part,( the OP level less than 28 mg kg-1) the crop yield showed a positive correlation with the increasing OP level; it is significantly related to the amount of P fertilizer. If the OP level was higher than 28 mg kg-1 there was no response to P fertilizer application, thus the crop yield was stable, despite the increase in the OP level and P application rates.

Line 299: This sentence is unclear

Author Response

Dear reviewer 1
Greetings

Please find my manuscript entitled “Effect of Soil Olsen P Levels on the Yield of Maize and Phosphorus Use Efficiency in Black Soil “number 1140847. I have revised it according to your comments and suggestions. The text in red color for comments and suggestions for reviewer 1, and in blue color for comments and suggestions for reviewer 2, as for the English editing, I  have sent the manuscript to MDPI, English editing services to revising the language. 
I hope this work will satisfy you.

Best wishes
Khalid Ibrahim

Reviewer 2 Report

I have read the manuscript ‚Effect of Soil Olsen P Levels on the Yield of Maize and Phosphorus Use Efficiency in Black Soil, submitted for review to Sustainability, carefully. The study investigated the effect of six phosphorus rates (0, 22, 44, 59, 73, and 177 kg P/ha, and six OP levels (16, 20,28,38,43, and 49 ppm) on yield parameters and other measurements. Data were collected during the years 2017, 2018, and 2019. The topic itself might be of interest to agronomists/farmers and scientists working in the agricultural scope. In my opinion, the text, however, misses out on large pieces of information that make the research understandable and/or reproducible. However, the obtained results are not satisfactory. Overall, this study did not have clear data to get enough results and a view to understanding the effects of P application and OP on Maize yield growth. However, I would like to see the ANOVA table or bar chart in the results that show the relations between P rates and yield, this could be good evidence to prove their results. Based on the current results which are not enough and clear to be understood. The introduction is not representing the main objectives of this study. The introduction should be written again to be more clear. The introduction should be changed to be more clear and sensible. The English language is the main issue here, authors should work on their English language and rewrite many sentences. Please see the attached pdf file which has all my questions, suggestions, and corrections.

Author Response

Dear reviewer 2
Greetings

Please find my manuscript entitled “Effect of Soil Olsen P Levels on the Yield of Maize and Phosphorus Use Efficiency in Black Soil “number 1140847. I have revised it according to your comments and suggestions. The text in blue color for comments and suggestions for reviewer 2, and in red color for comments and suggestions for reviewer 1, as for the English editing, I  have sent the manuscript to MDPI, English editing services to revising the language. 
I hope this work will satisfy you.

Best wishes
Khalid Ibrahim

Round 2

Reviewer 1 Report

Dear Authors,

the manuscript is now better written and more clearly. Please check the space between the lines in the whole manuscript, because in some paragraphs there is single line spacing and in other paragraphs the space between the lines is bigger.

Author Response

Dear reviewer

Greetings

Thank you so much for helping me to correct the manuscript. You find the attached file. According to your comment, I checked the space between the lines in the whole manuscript. The English language editing by MDPI was done.

I hope this work will satisfy you.

Kind regards

Dr. Khalid Ibrahim

Reviewer 2 Report

Please see the attached pdf file which has all my questions, suggestions, and corrections.

Author Response

Dear reviewer

Greetings

Thank you for helping me to correct the manuscript. You find the attached file of the manuscript; I revised it according to your comments and suggestions. The English language editing by MDPI was done.

I hope this work will satisfy you.

Kind regards

Dr. Khalid Ibrahim

Round 3

Reviewer 2 Report

There is some stuff that has to be done. I mentioned them in the attached pdf file. Then, It can be accepted.  

Congratulations

Author Response

Dear reviewer

Greetings

Thank you very much for your careful feedback on the manuscript. You find the attached file of the manuscript; I revised it according to your suggestions.

Kind regards

Dr Khalid Ibrahim
